# Potassium Spraying Preharvest and Nanocoating Postharvest Improve the Quality and Extend the Storage Period for Acid Lime (*Citrus aurantifolia* Swingle) Fruits

**DOI:** 10.3390/plants12223848

**Published:** 2023-11-14

**Authors:** Hamada R. Beheiry, Mohamed S. Hasanin, Amr Abdelkhalek, Hamdy A. Z. Hussein

**Affiliations:** 1Horticulture Department, Faculty of Agriculture, Fayoum University, Fayoum 63514, Egypt; hrh01@fayoum.edu.eg (H.R.B.); haz00@fayoum.edu.eg (H.A.Z.H.); 2Cellulose and Paper Department, National Research Centre, Dokki, Cairo 12622, Egypt; 3Horticultural Crops Technology Department, National Research Centre, Cairo 12622, Egypt

**Keywords:** nanocomposite, biopolymers, nanocoating, *Citrus aurantifolia* swingle, potassium spray

## Abstract

Citrus fruits are one of the most abundant crops globally in more than 140 countries throughout the world. Acid lime (*Citrus aurantifolia* swingle) is one of the citrus fruits which popularly has rich nutritional and therapeutic features. The storage period is the important factor that affects the economic and quality properties of this fruit. This study aims to demonstrate the enhancing effect of preharvest spraying with potassium, in addition to the postharvest dipping of fruits in some edible coatings, on the quality and storability of acid lime fruits. Preharvest spraying with organic and mineral forms of potassium, namely, potassium thiosulfate 1.75 g/L (S) and potassium tartrate 2 g/L (T), were carried out at three different times, in May, June, and July. On the other hand, postharvest treatments were carried out via dipping fruits in different types of biopolymers (carboxymethyl cellulose (E2) and gum arabic (E3)) and carboxymethyl cellulose/gum arabic composite (E4) as well as nanocoating formulation based on both biopolymers and doped zinc oxide nanoparticles (ZnONPs) (E1), which were prepared via acid lime peel waste extract. Herein, the physiochemical and morphological characterizations confirmed that the nanocoating was prepared at the nanoscale and doped with green synthesis ZnONPs, with recorded sizes of around 80 and 20 nm, respectively. Preharvest spraying with potassium tartrate enhanced fruit traits (Spraying with potassium tartrate at pre-harvest and nanocoating dipping at post-harvest (TE1), spraying with potassium tartrate at pre-harvest and carboxy methyl cellulose dipping at post-harvest (TE2), spraying with potassium tartrate at pre-harvest and gum arabic dipping at post-harvest (TE3) and spraying with potassium tartrate at pre-harvest and carboxymethyl cellulose/gum arabic composite dipping at post-harvest (TE4)), followed by potassium thiosulfate (spraying with potassium thiosulfate at pre-harvest and nanocoating dipping at post-harvest (SE1), spraying with potassium thiosulfate at pre-harvest and carboxy methyl cellulose dipping at post-harvest (SE2), spraying with potassium thiosulfate at pre-harvest and gum arabic dipping at post-harvest (SE3) and spraying with potassium thiosulfate at pre-harvest and carboxymethyl cellulose/gum arabic dipping at post-harvest (SE4)), compared to control. For postharvest treatments, E1 improved fruit quality, followed by E2, E4, and E3, respectively. The integration between pre- and postharvest treatments showed a clear superiority of TE2, followed by TE4, SE1, and SE2, respectively.

## 1. Introduction

Citrus is one of the most popular fruits in many countries around the world, including Egypt, which is one of the top ten citrus producers in the world [1]. Citrus fruits, like most other whole foods, contain a diverse range of essential nutrients, including glycemic and nonglycemic carbohydrates (sugars and fiber), potassium, folate, calcium, thiamin, niacin, vitamin B6, phosphorus, magnesium, copper, riboflavin, pantothenic acid, and a variety of phytochemicals [2]. Furthermore, citrus has no fat, sodium, or cholesterol because it is a plant food. Due to its high mineral and vitamin C content, acid lime (*Citrus aurantifolia* swingle), one of the citrus fruits, is a well-liked citrus fruit [3].

Potassium is a crucial nutrient in determining fruit size, productivity, and quality. Citrus is grown on sandy soils, which naturally contain little K. Also, even with repeated fertilizer treatments, K is not fixed and does not build in those sandy soils. Therefore, citrus orchards need to use K fertilizer. Potassium shortage is uncommon in groves that receive normal fertilization, although it can occur on high pH soils or when high N rates promote a high fruit production [4]. Potassium is necessary for several basic physiological functions such as forming sugars and starch, synthesizing proteins, normal cell division and growth, and neutralizing organic acids. Potassium reduces the influence of adverse weather conditions (such as drought, cold, and flooding) on citrus trees.

Fruit coating is a promising technique in which the fruit itself is coated with a layer of an active ingredient/s for preservation goals [5,6]. Indeed, the coating materials need to be biocompatible, nontoxic, not released, and sustainable to keep the price economical as well [7]. Edible coating materials based on biopolymers such as cellulose and its derivatives are favorable in such applications, as they have characteristics with all the features mentioned above in addition to sustainability and availability. Carboxymethyl cellulose (CMC), one of the cellulose ether derivatives, is also called cellulose gum and has characteristics with unique features such as a high viscosity, edibility, and a good dispersity [8]. CMC-based nanomaterials, especially nanometals, are widely used for several applications including medical, pharmaceutical, and packaging purposes [9]. In this context, CMC is used as a stabilized and dispersed agent for nanoparticle stabilization [10]. Gum arabic is a branched-chain, complex polysaccharide, either neutral or slightly acidic. The backbone is composed of 1,3-linked β-d-galactopyranosyl units. The side chains are composed of two to five 1,3-linked β-D-galactopyranosyl units, joined to the main chain by 1,6-linkages [11,12]. 

Acid lime (*Citrus aurantifolia* swingle) peel waste (LPW) is a zero-value waste produced from fruit processing, with a high content of many active ingredients including polyphenols, vitamins, minerals, and volatile oils [13,14]. LPW can be used in the biosynthesis of metal nanoparticles as a capping and stabilizing agent as well [15,16,17]. 

Zinc oxide nanoparticles (ZnONPs) are one of the safest and most biocompatible nanometals that perform with an excellent safety profile and can be formulated via a bioprocess including plant and microbial methods [18]. Moreover, the combination of polysaccharides and nanometals is a promising strategy to improve food packaging, especially fresh ones such as fruits and vegetables [19]. Therefore, in this presented work, a green ecofriendly method was used for the biosynthesis of ZnONPs using acid lime (*Citrus aurantifolia* Swingle) peel waste extract incorporated into a CMC matrix to formulate an edible nanocoating for acid lime (*Citrus aurantifolia* swingle) fruits. Additionally, the produced nanocoating films were characterized via their physicochemical, morphological, and antimicrobial activities and evaluated. Additionally, this study aims to demonstrate the effect of preharvest spraying with organic and mineral sources of potassium, in addition to postharvest dipping of fruits in some edible coatings on the quality and storability of acid lime (*Citrus aurantifolia* swingle) fruits.

## 2. Materials and Methods

### 2.1. Reagents, Chemicals and Microbial Media

Gum arabic (GA) was purchased from Sigma Aldrich, Burlington, MA, USA. Carboxymethyl cellulose was purchased from Fluka, Darmstadt, Germany, with the following specifications: H_2_O 5–10%; pH (1% in water) 6.5–7.5; purity was more than 99.5% and the average degree of substitution (DS) was 0.79. Zinc acetate was purchased from Sigma Chemical Co., Ltd. (St. Louis, MO, USA). All microbial media and reagents were purchased from Loba Chem., India in analytical grade.

### 2.2. Fruits

The present investigation was carried out during the growing season of 2021/2022 on acid lime (*Citrus aurantfolia* swingle) Balady cv in a private orchard in the Fedemin region (about 10 km west of Fayoum), Fayoum Governorate, Egypt (29.41° N, 30.767° E) on ten-year-old trees at the beginning of the study.

### 2.3. Treatments

#### 2.3.1. Preharvest Treatment

Trees spaced 5 m × 5 m apart, (400 trees/hectares). The experiment comprised three treatments of K fertilizers and their rate of application, viz., potassium thiosulphate at 1.75 g/L (S) and potassium tartrate at 2 g/L (S), which were compared with a control, i.e., trees sprayed with distilled water.

There were three sprayings in the last week of May, June, and July. The foliar application was done by using the rate of 20 L/tree. All nine treatments were replicated three times, taking one tree as a single unit. Twenty-seven lime trees having a uniform size and vigor were selected for the investigation. Cultural practices such as irrigation, fertilization, and plant protection were kept uniform for all treatments.

Soil samples were collected randomly from a 40 cm deep core for nutrient analysis and physical and chemical characteristics (Table 1) according to the methods developed by Bortolon et al. (2011) [20].

Moreover, meteorological data for the same period in the same suite were also documented (Figure 1).

All trees were also fertilized with triple superphosphate at 3 kg/tree and ammonium sulfate at 3 kg tree^−1^ according to the recommendation from the Ministry of Agriculture and Land Reclamation, Egypt.

#### 2.3.2. Postharvest Treatments

Fruits were harvested in the first week of September. Data were collected in fresh conditions and then at 15-day intervals up to 60 days. Data from four fruits of each replicate were recorded. 

The following treatments were carried out on fruits via a nanocoating matrix that was prepared via a green method as follows.

### 2.4. Preparation of Nanocoating

The nanocoating used to dope ZnONPs included GA and CMC, which were prepared using acid lime (*Citrus aurantifolia* swingle) peel waste.

#### 2.4.1. Biosynthesis of ZnONPs

Acid lime (*Citrus aurantifolia* swingle) peel wastes resulted from acid lime fruit processing which was collected from the free markets. Bright LPW that appeared free of any disease symptoms was selected and washed thoroughly with distilled water. The alcoholic extraction (70%) as 10 g of LPW per 100 mL was ultrasonicated in a water bath for 1 h at 60 °C. Afterward, the total extract was filtrated and subjected to the biosynthesis of ZnONPs. Briefly, 100 mL of total extract was stirred at 1500 rpm after adding 5 g of zinc acetate. This mixture was heated at 70 °C for 2 h and the collected mixture was ultrasonicated via an ultrasonic probe for 5 min. The filtrated solution was lyophilized for further use.

#### 2.4.2. Nanocoating Formulation

Nanocoating films were prepared via a casting method using 50 mL of a CMC (1% *w*/*v*) aqueous solution and 50 mL of a GA solution (1% *w*/*v*). Both solutions were mixed with stirring at 70 °C for 1 h until complete mixing. ZnONPs were added in a fixed mass of 2 g as a safety percentage for edibility [21]. The above-prepared solution was stirred in the same condition for 1 h and ultrasonicated via an ultrasonic probe for 5 min. The collected solution was preserved in a refrigerator for further use.

#### 2.4.3. Nanocoating Characterizations

The prepared nanocoating and its raw materials, as well as the prepared ZnONPs, were characterized using spectroscopy techniques. UV–visible spectra were measured using a UV–vis spectrophotometer (Jasco, V-630, Tokyo, Japan) in the range of 200–1000 nm. FT-IR spectra were measured by ATR-FTIR spectroscopy (Spectrum Two IR Spectrometer—PerkinElmer, Inc., Shelton, CT, USA). All spectra were obtained by 32 scans and a 4 cm^−1^ resolution in wavenumbers ranging from 4000 to 400 cm^−1^. An XRD analysis was carried out at a 2θ of 5–80°, using a Bruker D8 Advance X-ray diffractometer (Mannheim, Germany). Using field emission SEM in conjunction with an energy-dispersive X-ray analysis, a Model Quanta 250 FEG (field emission gun) attached to an EDX Unit (energy-dispersive X-ray analyses) was used to determine the topography as well as particle size.

### 2.5. Fruit Analyses

#### 2.5.1. Fruit Decay

Fruit decay was recorded at 15-day intervals and the percentage was calculated from the total number of fruits.

#### 2.5.2. Physical Characteristics

The following data were recorded during the experimental period at harvest and every 15 days up to 60 days.

##### Fruit Weight

The weight of each fruit at the time of cold storage was taken in grams using an electric balance and was recorded. Afterward, at 15-day intervals, fruit weight in grams was also recorded.

##### Fruit Length and Fruit Diameter

Fruit length and fruit diameter (mm) were recorded with digital Vernier’s calipers, and the average value was expressed in millimeters (mm).

##### Fruit Volume

The volume of each lime fruit was measured using the water displacement method. Each fruit was submerged in a container full of water, and the volume of displaced water was directly measured using a 250 cm^3^ graduated cylinder. Water temperature during measurements was maintained at 25 °C.

##### Fruit Firmness

It was measured with the help of a penetrometer. A small section of fruit peel was removed, the penetrometer nob was inserted gently into the fruit, and the value was recorded. The process was repeated three times, and the average was taken.

##### Fruit Juice Volume

Juice volume was determined by squeezing the fruit and estimating the volume of produced juice in cm^3^.

#### 2.5.3. Fruit Chemical Characteristics

For the measurement of fruit chemical parameters from each replicate in each treatment, 100 fruits were harvested. For each treatment (*n* = 9), three juice samples were used for the chemical analysis. A commercial manual juicer was used to precisely extract lime fruit juice at constant room temperature.

Immediately after being squeezed, the juice was characterized for total soluble solids (TSS, °Brix), titratable acidity (TA, g acid citric L^−1^), and TSS:TA ratio, and the vitamin C content was also calculated as in the following.

##### Total Soluble Solids

The total soluble solids (TSS) content of acid lime pulp was estimated by using a hand refractometer. One drop of acid lime juice was squeezed from the fruit pulp on the prism of the refractometer (model no. ATAGONA-Brix 0-32) and the percentage of total soluble solids was obtained from a direct reading.

##### Titratable Acidity

In a 250 mL beaker, 10 mL of the juice extract was mixed with 20 mL of distilled water. Juice samples were titrated against 0.01 N NaOH, and the reading was recorded when the color changed to light pink. Titratable acidity (TA) was determined by the standard method of Cunniff, P. and Washington, D., 1997 [22].

##### Ascorbic Acid Content

The standard method of Cunniff, P. and Washington, D., 1997 [22] was used to determine the ascorbic acid of lime fruits. The titration was started against a dye, and the reading was recorded when the sample changed into a light pink color

#### 2.5.4. Fruit Weight Loss

Fruit weight loss in percentage was calculated from the initial weight and the weight was taken at 15-day intervals.

### 2.6. Statistical Analysis

An ANOVA of the data was conducted using the InfoStat statistical package, version 2011; the replicate was considered the random variable. This means multiple comparisons were conducted using the Duncan test at *p* ≤ 0.05. A statistical analysis of the data was carried out according to Snedecor and Cochran (1994) [23].

## 3. Results

### 3.1. Nanocoating Characterizations

#### 3.1.1. Physicochemical Analysis

The nanocoating was characterized and compared with its neat materials using physicochemical and topographical analyses. The physicochemical analysis, including UV–visible spectroscopy shown in Figure 2, illustrated the formulation of nanosized ZnONPs with a characteristic band at 364 nm that shifted to 330 nm after being incorporated into the nanocoating.

In this context, Figure 3 presents the FTIR spectra of the prepared nanocoating and its neat components. Sample E2 showed characteristic bands that were assigned to 3294, 2868, 1585, 1410, 1321, and 1020 cm^−1^ corresponding to hydroxyl groups’ stretching vibrations, CH stretching, C=O groups, aliphatic CH stretching vibration from the carboxyl group (COO-), carboxyl group as it salts (COO-Na), and glycosidic linkage bond, respectively [9]. Sample E3’s spectrum exhibited characteristic bands at 3301, 2931, 1600, 1413, and 1020 cm^−1^ referring to hydroxyl groups stretching vibrations, CH stretching, C=O groups, aliphatic CH, and glycosidic linkage bond, respectively [12]. Otherwise, the sample E4’s bands at 3290, 2915, 1598, 1415, 1321, and 1010 cm^−1^, as well as the sample E1’s presented bands at 3310, 2924, 1585, 1405, 1327, and 1014 cm^−1^ represented the polysaccharide skeleton. Additionally, the characteristic bands of the stretching vibration of the Zn-O bond were from approximately 410 to 488 cm ^−1^ [19].

Figure 4 presents the XRD pattern of the formulated nanocoating and neat materials. The E3 pattern exhibited peaks at 18.7° and 42.3°, predictable for amorphous gum arabic [23]. The E2 pattern was observed in two hub peaks at around 10° and 19.5° that referred to the amorphous structure of CMC [24]. The E4 pattern was observed as a low crystalline pattern in comparison with neat materials, and the peaks were recorded with a low intensity compared with neat materials as well. In addition, the XRD confirmed the characteristic patterns of prepared ZnONPs at 32, 36, 38, 48, 57, 64, and 67°, which was attributed to (100), (002), (101), (102), (110), (103), and (112) crystal planes, respectively, and fully matched with the card wurtzite crystal structure (JCPDS card No. 36-1451) [25]. On the other hand, the E1 samples were observed as a crystalline pattern in comparison with E4’s significant sharp peaks at 30, 36, and 48°.

#### 3.1.2. Topography

A topographical analysis was carried out to assess the morphological performance of the prepared samples. SEM and EDX are presented in Figure 5 with the simulation of the surface roughness. The E2 sample’s SEM image showed cellulosic fibers compacted together with a rough surface morphology, and the EDX chart displayed carbon, oxygen, and sodium ions. In addition, the E3 sample’s SEM image exhibited layer sheets with smooth surfaces in comparison with sample E2, with an EDX chart showing carbon, oxygen, and nitrogen atoms. On the other hand, sample E4 had a new morphological surface that was different compared to both neat materials, and the EDX chart recorded a sum of both elements recorded in samples E2 and E3. However, sample E1 was similar to sample E4 with some metallic chains on the surface, and the EDX chart presented zinc ions as well.

Figure 6 presents the TEM images of ZnONPs with low and high magnifications, as well as the SEAD pattern showing a nanoscale of ZnO particles with a size of around 20 nm, and the SEAD pattern was a crystalline pattern. Additionally, sample E1 formed a nanonetwork with obvious nanoparticles on the nonmatrix structure with a size of around 80 nm, while the SEAD pattern was observed as a polycrystalline pattern with about four faint rings.

### 3.2. Postharvest Fruit Characteristics

#### 3.2.1. Fruit Decay

Regarding the effect of preharvest treatments on fruit decay percentage (Table 2), there was a clear superiority of preharvest treatments (potassium thiosulfate and potassium tartrate) compared to control, as the percentage of decay was the least possible when spraying before harvest with potassium thiosulfate, followed by a significant difference by spraying with potassium tartrate, which gave a higher percentage of fruit decay. As for the postharvest dipping treatments, they also had a clear effect on fruit decay percentage, as potassium thiosulfate with gum arabic (SE3) achieved the lowest percentage of decay, followed by potassium thiosulfate with carboxymethyl cellulose (SE2), while the control gave the highest rate of fruit decay.

With regard to the effect of the postharvest cold storage period on all treatments, the fruits were not affected in the first fifteen days of cold storage, but after a month of storage, some fruits began to be affected, and there was a direct proportion between the storage period and percentage of fruit decay until it reached the highest percentage of decay after sixty days of cold storage.

According to the interaction between pre- and postharvest treatments and the storage period, the lowest percentage of fruit decay after the first fifteen days of cold storage was for potassium thiosulfate and gum arabic (SE3), followed by potassium thiosulfate and carboxymethyl cellulose (SE2) for thirty days, respectively. However, the highest percentage of decay appeared in the S control and control treatments for 60 days, respectively.

#### 3.2.2. Fruit Physical Characteristics

##### Fruit Length and Fruit Diameter

Preharvest treatments had the highest and most obvious effect on fruit length more than other treatments (Table 3). Potassium thiosulfate achieved the highest fruit height, followed by potassium tartrate, then control. However, throughout the results, it was clear that the highest average fruit length appeared when spraying before harvest with potassium thiosulfate and dipping the fruits after harvest in gum arabic extract (SE3), while the lowest length of fruit was evident with the control.

There was an inverse proportion between average fruit length and storage periods, where the length of fruit at time zero was 3.86, and it continued to decrease until it reached 3.39 at 60 days of cold storage.

Through consolidating all treatments and periods, SE3 for 15 days had the highest average fruit length (4.13 cm), while the control treatment for 60 days had the lowest (3.39 cm).

The highest value of average fruit diameter appeared when potassium thiosulfate 1.75 cm^3^/L (preharvest treatment) + CMC and a GA mixture (postharvest treatment) (TE4) treatment was used (3.81 cm), followed by insignificant differences in both SE2 and SE3, which had the same value (3.61 cm), while the lowest value appeared with the control (3.1 cm).

The diameter of the fruit also decreased over time during storage, as did its length, and this is evident in Figure 7.

TE4 at time zero was superior in average fruit diameter (4.07 cm), while the control treatment for 60 days (2.77 cm) produced the lowest diameters, and this clearly shows the effect of preharvest spraying treatments on fruit diameter.

##### Fruit Volume and Fruit Weight

Concerning fruit volume, it was found that SE2 had the highest value, followed by SE3, although there was a minor difference between them, while the control had the lowest value among the rest of all treatments. The longer the storage period, the lower the volume of fruits, and this is evident from the Figure 8. Although there were no clear differences between the treatments and storage period, the highest value for fruit volume appeared in treatment SE2 for 15 days of cold storage, and the lowest value came from the control treatment for 60 days.

It was observed that fruit weight followed the same line as fruit volume, as the highest value emerged in the SE2 treatment and the lowest value in the control, as shown in Table 3. In relation to the storage period, the value of the fruit weight was highest at time zero and lowest after a period of 60 days, as seen in Table 4. About the interplay between treatments and cold storage periods, SE2 for 15 days had the highest fruit volume, followed by a negligible value for SE2 at time zero, while the lowest value was seen in the control treatment for 60 days.

##### Fruit Firmness

All treatments (pre- and postharvest), ignoring the storage period, showed a clear superiority in fruit firmness compared to the control treatments (S control, T control, and control) Figure 9.

Like all previous characteristics, fruit firmness decreased over time to a minimum at 60 days of cold storage.

Potassium thiosulfate with carboxymethyl cellulose (SE2) at zero time of storage had the highest value in fruit firmness (kg cm^−2^). On the contrary, T control for 60 days had the lowest.

##### Fruit Juice Volume

There were no significant differences between SE2 and SE3 regarding fruit juice volume, although SE3 was superior. The lowest volume of fruit juice was obtained in the control treatment (Table 5).

A decrease in juice volume was observed with a prolonged storage period after harvest, up to 60 days of cold storage of fruits, in which the volume of juice was the lowest.

Concerning the interconnection between treatments and the storage period, it was observed that SE2 at time zero had the highest fruit juice volume, followed by SE3 for 15 days and zero time of cold storage, respectively.

#### 3.2.3. Fruit Chemical characteristics

##### Total Soluble Solids and Titratable Acidity

The highest value for TSS was obtained in TE4, but the control treatment had the lowest value (Table 6). The TSS took an opposite path concerning the cold storage period, as the highest value for TSS appeared at 60 days of storage, although there were no clear significant differences.

According to the relatedness between treatments and storage period, it was observed that TE3 and TE4 for 45 and 60 days of cold storage were superior in TSS. On the contrary, the lowest values appeared in the control treatment at 30 and 60 days of cold storage, respectively.

SE1 and TE1 achieved the lowest levels of acidity, although there were significant differences between all S and T treatments, but the S control, T control, and control treatments showed the highest percentage of fruit juice acidity, as seen in Table 7.

Regarding the storage period, the acidity decreased over time until it reached its lowest level after 60 days of cold storage.

According to the correlation between treatments and the storage period, it was noted that TE2 for 60 days of cold storage was the lowest in terms of acidity, followed by TE1 for 45 days, while the S control, T control, and control treatments for 15 days of cold storage gave the highest fruit juice acidity.

##### TSS/Acidity

For the TSS/acidity, it was found that TE2 produced the highest score, followed by TE1, while the control treatment obtained the lowest rate of TSS/acidity, followed by T control and S control, respectively, as shown in Figure 10. According to the storage period, the highest ratio of TSS/acidity was acquired after 60 days of cold storage, while the lowest percentage was at 30 days. Interlinkages between treatments and storage period showed unclear differences, but they tended to deal with TE2 for 60 days and TE1 for 45 days, respectively, while the lowest TSS/acidity was obtained in the control treatment for 15 days and at time zero, respectively.

##### Vitamin C Content

All pre- and postharvest treatments showed a clear superiority with regard to the percentage of vitamin C compared to all control treatments (Table 8). The vitamin C content of fruits decreased over time during cold storage, where the highest percentage was at time zero, and the lowest was at 60 days. TE3 for 15 days and SE1 for 30 days attained the highest vitamin C content, respectively, while the lowest content was for the control treatment at 60 days of cold storage.

### 3.3. Fruit Weight Loss

The lowest percentage of fruit weight loss was found in treatment TE2, followed by TE3, while the highest percentages appeared in all control treatments (S, T, and control), as shown in Figure 11. Fruit weight loss increased over time during postharvest cold storage, until it reached the highest percentage at 60 days of storage. The interactivity between treatments and storage period showed that TE3 for 15 days achieved the lowest percentage of fruit weight loss, followed by SE4 and TE4 for 15 days, respectively, but the highest percentage of fruit weight loss with a significant increase resulted from the control treatment, then the S control and T control treatments for 60 days of cold storage, respectively.

## 4. Discussion

### 4.1. Nanocoating Characterizations

#### 4.1.1. Physicochemical Analysis

The physicochemical analysis involved UV–vis, FTIR, and XRD analyses which characterized the nanocoating and its neat materials. UV–vis (Figure 2) spectra affirmed the biosynthesis of ZnONPs and recorded a characteristic band at 364 nm, which is in nice agreement with our previous work [7]. Moreover, this band was affected after the incorporation of ZnONPs into the nanocoating matrix (E1), which shifted it to 330 nm. In addition, the FTIR spectra (Figure 3) of samples E3, E2, E4, and E1 confirmed the main changes between the nanocoating and neat materials. The E4 sample spectrum obviously presented a significant interaction between the E2 and E3 materials’ networks with a significant shift of the main bands in comparison with the neat spectrum of each material. In addition, sample E1 presented a significant shift in the characteristic band of sample E4 due to the interaction between the chains with ZnONPs. Moreover, the XRD pattern of the samples (Figure 4) was observed as a significant change in the band positions and intensity, while sample E1 exhibited an interaction between the matrix of nanocoating and ZnONPs while maintaining the main characteristics bands of ZnONPs.

#### 4.1.2. Topographical Analysis

The SEM and EDX analyses and the simulation of surface roughness were presented in Figure 5. Sample E2 presented cellulosic fibers that were in nice agreement with the cellulose derivative SEM image, which also exhibited a surface roughness with a high rough surface, and the EDX chart indicated a main composition of CMC. In addition, the E3 sample displayed as layers with more smoothness in comparison with sample E2, and the EDX chart presented a tical chemical composition of GA. However, sample E4 exhibited fine fibers with a smooth surface that referred to the interaction between the chains of samples E2 and E3, and the EDX chart presented all the elements of both samples, including carbon, oxygen, nitrogen, and sodium. In this context, sample E1 appeared close to sample E4 with some aggregations due to the presence of ZnONPs.

TEM images (Figure 6) presented the nanostructure of ZnO particles with a size of around 20 nm, and the nanocoating (E1) displayed nanofibers interacting together and doped ZnONPs. In addition, the SEAD pattern of both ZnONPs and sample E1 were noticed as crystalline and polycrystalline, respectively. Overall, the characterization techniques confirmed that both ZnONPs and nanocoating were prepared in the nanostructure.

It is well known that K is an element of fruit quality. This may be due to its functions in regulating the synthesis and transport of carbohydrates and activating enzyme activities [26,27]. Next is an illustration of how potassium affects the physicochemical characteristics of fruits.

### 4.2. Fruit Physical Characteristics

According to Vijay et al. (2017) [28], potassium treatment increases the weight of sweet orange fruit, which may be related to improved photosynthesis that produces additional carbohydrates. Another probable cause could be the greater mobility of assimilates by potassium to the developing fruits, which acts as a strong metabolic sink. The results showed that all potassium treatments increased fruit size, as measured by fruit diameter and the proportion of medium and big fruits compared to the control treatment, and that fruit size increased as K dosages increased, similar to how the size of Florida citrus fruits increases when increasing the rate and frequency of a K foliar treatment. The significance of potassium in the synthesis of cell walls may be the cause of the larger fruit size.

The availability of potassium may have boosted the stream of sucrose to the apoplast, resulting in a greater sugar transfer to the sink tissues and promoting fruit development, which may explain the rise in fruit weight. Potassium regulates the opening and closing of stomata in leaves, which increases the water absorption capacity of the plants, and as a result, fruit firmness increases. Potassium application speeds up the osmoregulation of cell vacuoles and maintains balance, producing hard fruits that are in nice agreement with Sajid et al. (2022) [29].

Hamza et al. (2012) [30] also observed that the treatment of 8% KNO_3_ (sprayed thrice) gave the best percentage of extra size class (57–63 mm) in citrus Clementine var. *Cadoux*. Yadav et al. (2014) [31] also reported a significant increase in fruit diameter with three sprays of 2% K_2_SO_4_ in ber fruits.

### 4.3. Fruit Chemical Characteristics

Our results are indicative of the elevating effect of potassium application (spraying acid lime (*Citrus aurantifolia)* trees with potassium tartrate (2 g/L) and potassium thiosulfate (1.75 cm^3^/L) three times before harvesting) on fruit TSS, which is in agreement with Alva et al.’s (2006) [32] findings. Also, potassium treatment increased tomato TSS [33]. Improvement in fruit quality can be attributed to the effects of potassium on carbohydrates and/or PGRs in developing fruit.

As a powerful antioxidant, ascorbic acid is one of the most important quality characteristics of fruits and vegetables [34]. Ascorbic acid concentration is influenced by species, the stage of ripening at the harvest, and pre- and postharvest treatments, such as the foliar application of micro- and macronutrients and PGRs [35]. Potassium had a significant effect on ascorbic acid content, and these results are in line with Alva et al. (2006) [32]. As mentioned earlier, potassium is an essential element for the growth of fruit cells, balancing the carbohydrates and the maintenance of quality [36].

Ascorbic acid levels were greater in potassium nitrate preharvest-treated pear fruits than in the control treatment. Bhat et al. (2012) [37] also noted an increase in ascorbic acid concentration in cherry with potassium nitrate foliation.

The improvement in TSS content with the foliar application of potassium might be due to the fact that potassium enhances the translocation of sugars from leaves to fruits. Potassium foliar applications have an important role in nutrient translocation from source to sink, which results in better-quality produce having a maximum quantity of TSS [29].

Potassium plays a crucial function in water intake and consumption, as well as in greater nutrition water uptake that increases juice content. It also aids in the translocation of sugar and is employed as an osmosis agent to open and close stomata. Similar results were found in citrus by Alva et al. (2006) [32], who reported that the high juice content may be related to potassium’s regulatory role in many physiological and biochemical processes of the plant, including the activation of enzymes, protein synthesis, stomatal function, stabilization of internal pH, photosynthesis, turgor-related processes, transport of metabolites, and extensibility. The investigation’s findings, as stated above, closely align with that of Kumar et al. (2015) [38] in guava, who reported that the high TSS content might be because of the translocation of photoassimilates, sugars, and other soluble solids that lead to a rise in TSS, in which potassium plays a significant role. Bhavya et al. (2010) [39] also reported that the increment of TSS in grapes leads to a decrease in acidity content, whereas potassium neutralizes organic acids and plays a role in controlling the acidity and pH of fruit juice. Vijay et al. (2016) [40] also reported in sweet orange that high sugar content minimized the acidity in fruit. In other words, the reduction in acidity might be due to an increase in the TSS of fruits.

Fruits’ titratable acidity constantly trended downward as more KNO_3_ was applied. The high TSS of the fruits might be responsible for the decrease in acidity after potassium treatment [41].

K is a key nutrient that plays a role in the photosynthetic and metabolic processes in plants and a key role in the taste and flavor of fruits [42]. The application of K increased the enzyme activities of sugar metabolism in fruits [43] and stimulated the transport of nutrients and sugar [44], leading to increased TSS content. K applied via foliar fertilization enhanced the TSS, decreased juice acidity, and improved fruit yield [45].

Potassium has a very vital role in the refulgence of biotic stresses, which damage the plants and increase fruit decay, as it plays a key role in the synthesis of high-molecular-weight compounds such as proteins, starch, and cellulose and reduces the collection of soluble sugars, organic acids, and amides on which pathogens feed [29].

## 5. Conclusions

The enhancement of strategic crops is a main issue that has attracted the attention of researchers for many years. In this work, nanocoating-based ecofriendly components prepared via a green method were used to treat acid lime (*Citrus aurantifolia* swingle) fruits. Herein, the biosynthesis of ZnONPs was carried out from LPW and presented a promising ability to reduce zinc to nanoform. In addition, the prepared nanocoating was observed to have unique physicochemical and morphological characteristics. Preharvest spraying with potassium tartrate enhanced fruit traits, followed by potassium thiosulfate, compared to the control treatment. For postharvest treatments, E1 improved fruit quality, followed by E2, E4, and E3, respectively. The integration between pre- and postharvest treatments showed a clear superiority of TE2, followed by TE4, SE1, and SE2, respectively.

## Figures and Tables

**Figure 1 plants-12-03848-f001:**
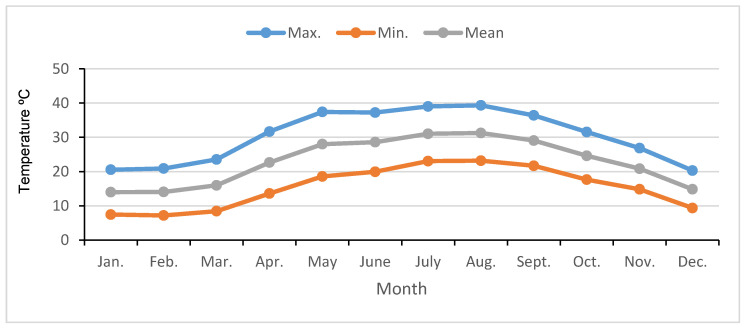
Monthly means of maximum, minimum, and mean temperature of years (°C) during the 2021/2022 season under Fayoum climatic conditions.

**Figure 2 plants-12-03848-f002:**
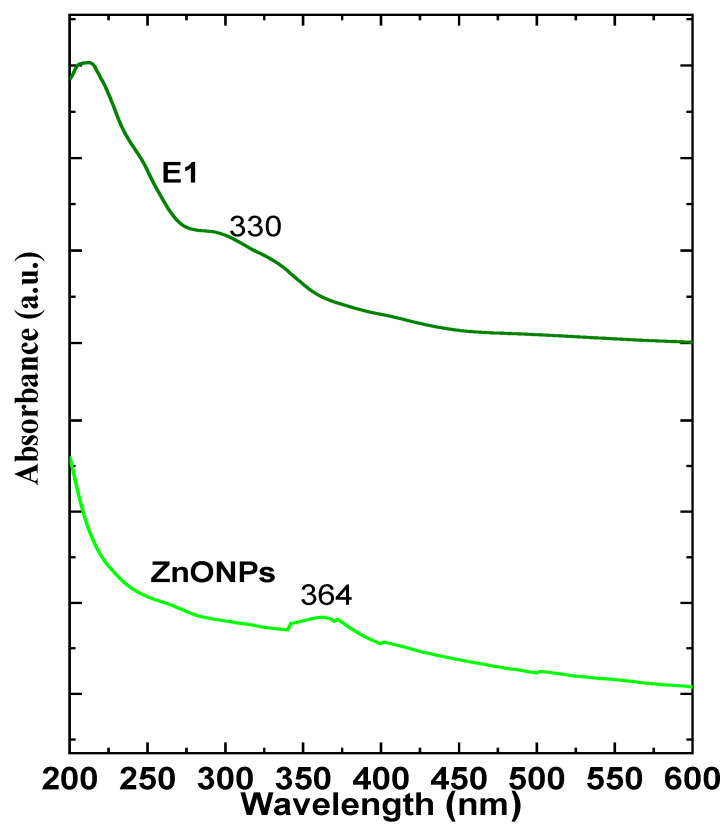
UV spectra of ZnONPs and nanocoating formulations.

**Figure 3 plants-12-03848-f003:**
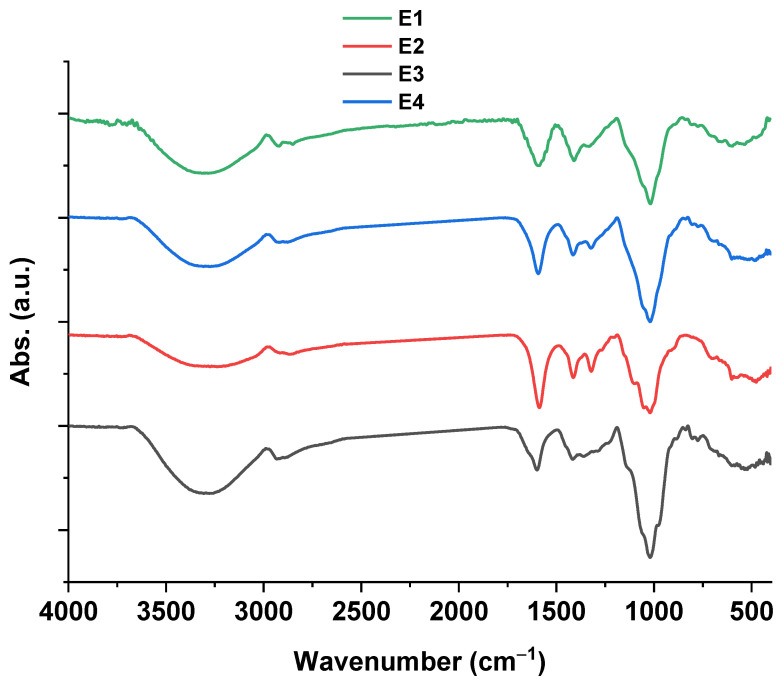
FTIR spectra of ZnONPs and E1 formulation as well as its neat materials.

**Figure 4 plants-12-03848-f004:**
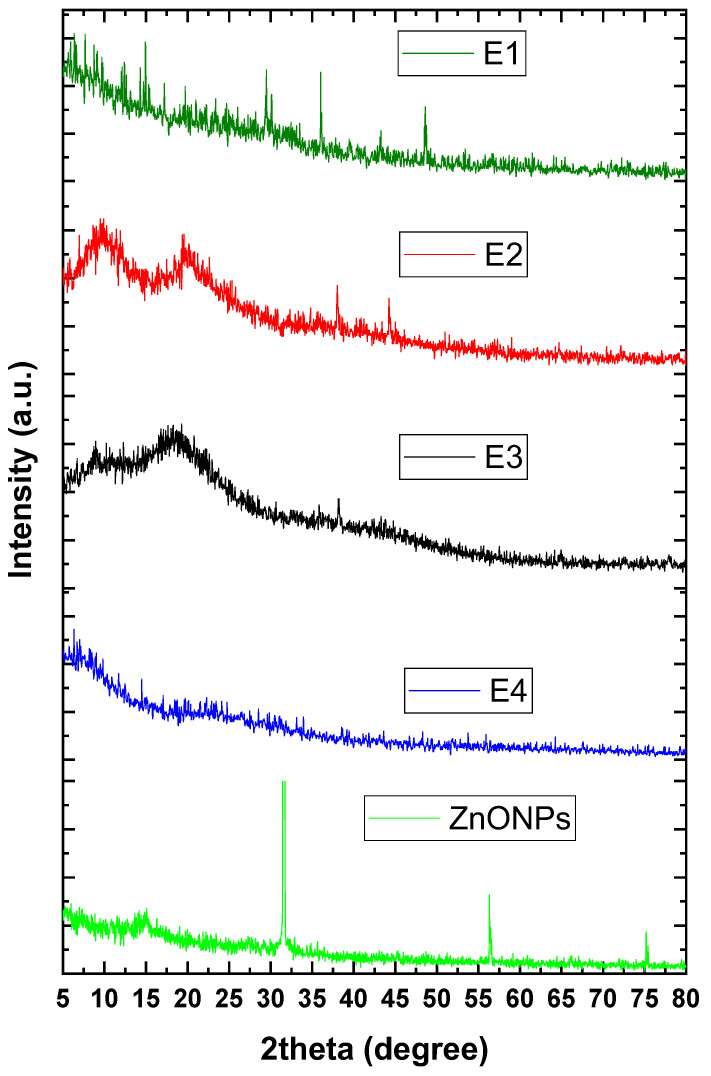
XRD patterns of ZnONPs and nanocoating formulation, as well as its neat materials.

**Figure 5 plants-12-03848-f005:**
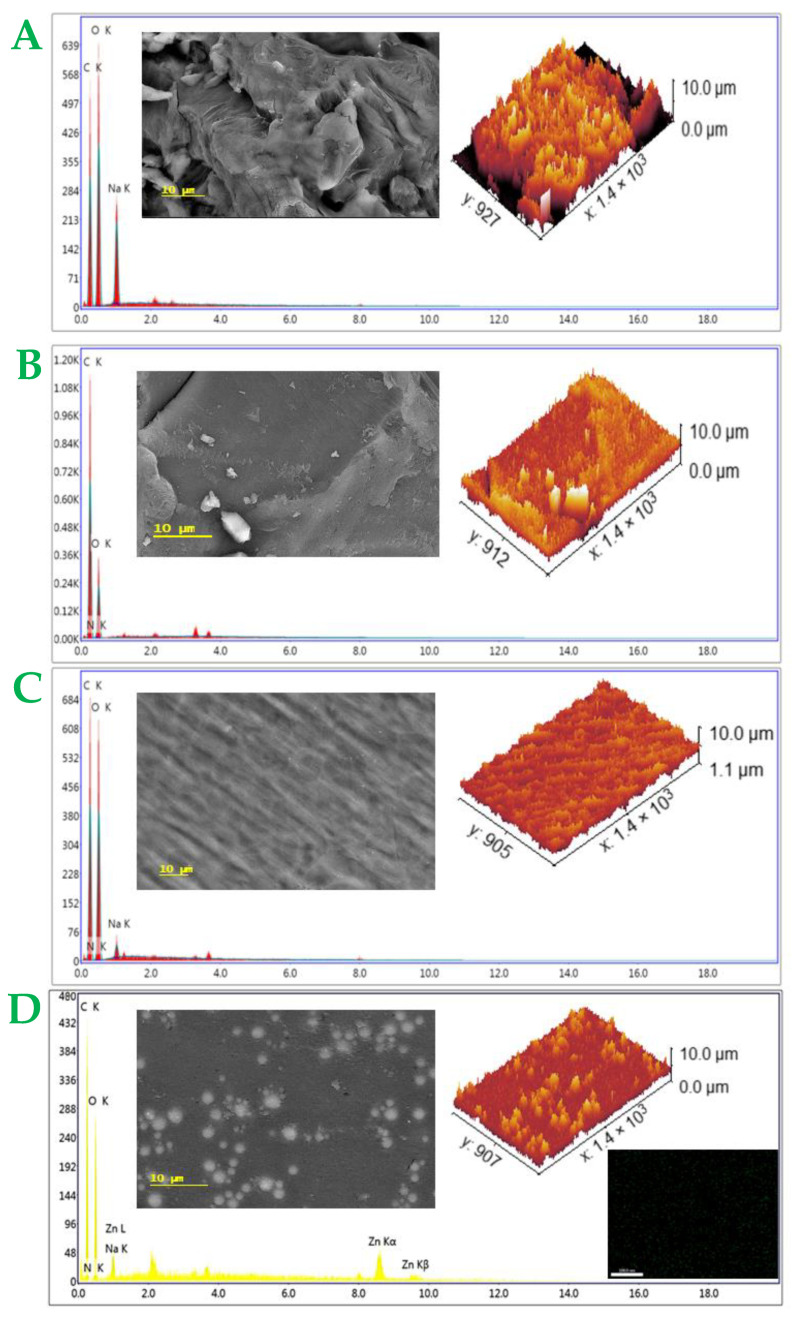
SEM images, EDX, and surface morphology simulation of E2 (**A**), E3 (**B**), E4 (**C**), and E1 (**D**).

**Figure 6 plants-12-03848-f006:**
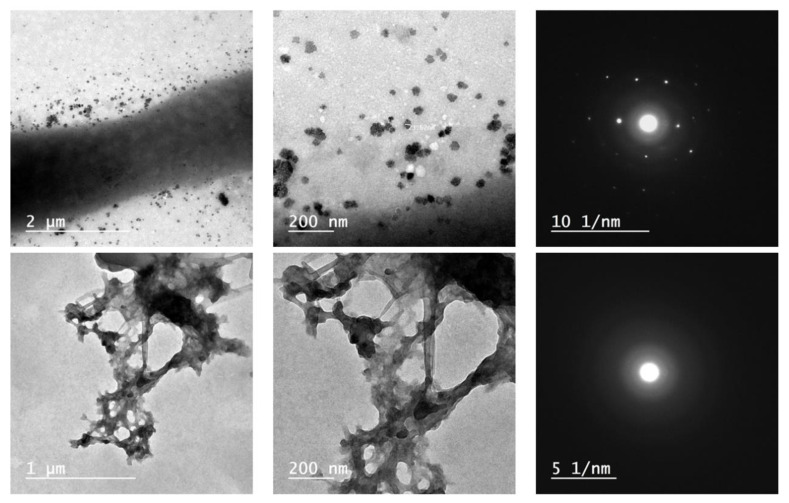
TEM images of ZnONPs (**upper**) and E1 (**lower**), as well as SEAD pattern.

**Figure 7 plants-12-03848-f007:**
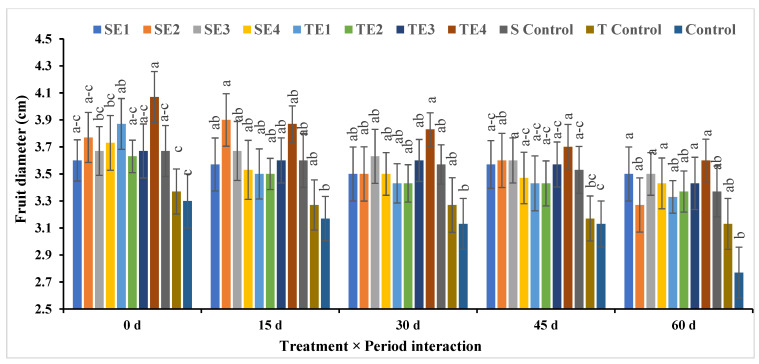
Effect of pre- and postharvest treatment on the fruit diameter (cm) of acid lime (*Citrus aurantifolia* S.) fruits. Mean values (±SE) with different letters in each bar are significant (at *p* ≤ 0.05). The same letters are not significantly different (at *p* ≤ 0.05).

**Figure 8 plants-12-03848-f008:**
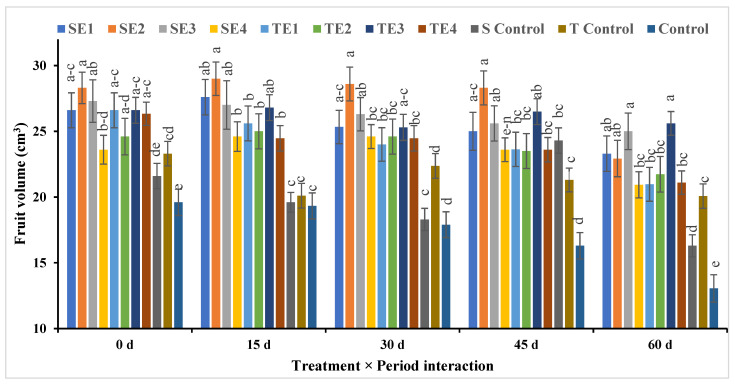
Effect of pre- and postharvest treatment on the fruit volume (cm^3^) of acid lime (*Citrus aurantifolia* S.) fruit. Mean values (±SE) with different letters in each bar are significant (at *p* ≤ 0.05). The same letters are not significantly different (at *p* ≤ 0.05).

**Figure 9 plants-12-03848-f009:**
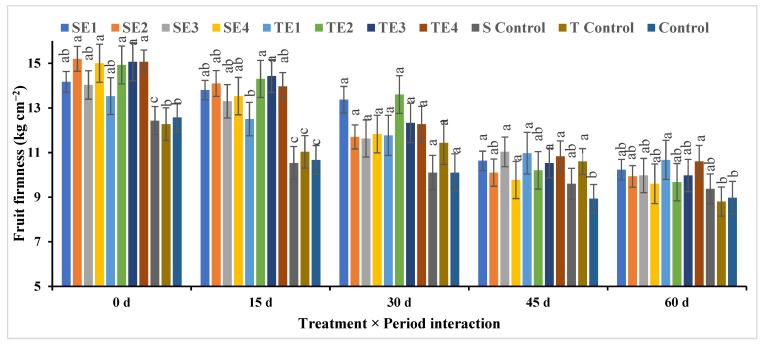
Effect of pre- and postharvest treatment on fruit firmness (kg.cm^−2^) of acid lime (*Citrus aurantifolia*). Mean values (±SE) with the same letters in each bar are not significantly different (at *p* ≤ 0.05.

**Figure 10 plants-12-03848-f010:**
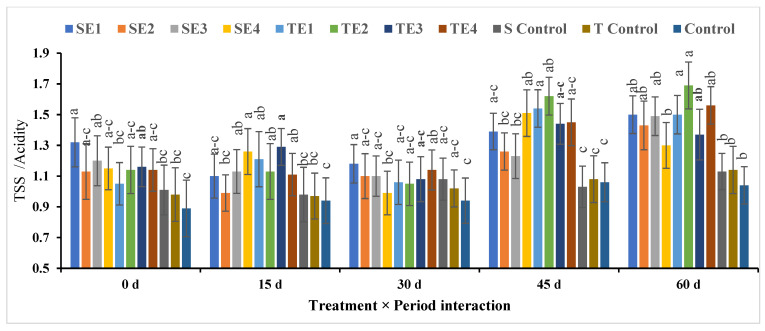
Effect of pre- and postharvest treatment on the TSS/acidity of acid lime (*Citrus aurantifolia* S.) fruit. Mean values (±SE) with the same letters in each bar are not significantly different (at *p* ≤ 0.05).

**Figure 11 plants-12-03848-f011:**
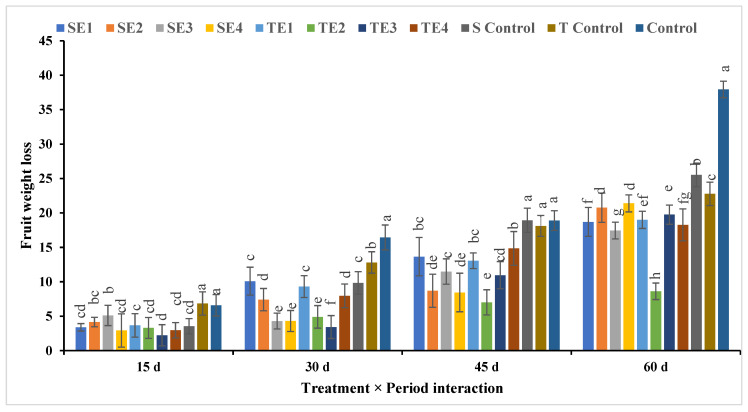
Effect of pre- and postharvest treatment on the fruit weight loss of acid lime (*Citrus aurantifolia* S.) fruit. Mean values (±SE) with different letters in each bar are significant (at *p* ≤ 0.05). The same letters are not significantly different (at *p* ≤ 0.05).

**Table 1 plants-12-03848-t001:** Physical and chemical characteristics of the soil at 0–40 cm depth on Balady lime (*Citrus aurantifolia* swingle) during the 2021/2022 season.

Chemical Characteristics													Physical Characteristics	
pH	Ece (ds/m)	Ca^2+^	Mg^2+^	Na^+^	K^+^	Co_3_^−^	HCO_3_^−^	Cl^−^	SO_4_^−^	Fe	Zn	Mn	Clay, %	65.20
In soil paste		Soluble cations, SC (meq/L)				Soluble anions, SA (meq/L)				DTPA extractable (mg/kg soil)			Silt, %	13.50
													Sand, %	21.40
7.46	3.08	10.84	1.25	6.03	1.23	1.70	1.20	6.46	9.08	76.0	21.0	27.2	Texture	clay

**Table 2 plants-12-03848-t002:** Effect of pre- and postharvest treatments on the decay (%) of acid lime (*Citrus aurantifolia* S.) fruits.

	Period (P; Days)					
Treatment	0	15	30	45	60	Mean (Treat.)
SE1	0.00	0.00	5.15ef ± 0.91	10.91f ± 1.21	30.44c ± 1.11	9.30G ± 3.84
SE2	0.00	0.00	3.74fg ± 0.53	10.48f ± 1.71	20.89e ± 1.86	7.02HI ± 2.55
SE3	0.00	0.00	1.65g ± 0.31	10.27f ± 0.45	21.64e ± 1.39	6.71I ± 2.93
SE4	0.00	0.00	6.21def ± 0.60	15.10de ± 0.97	18.77e ± 1.60	8.01H ± 4.55
TE1	0.00	0.00	6.42def ± 0.41	14.37e ± 0.63	26.60d ± 1.60	9.48G ± 3.89
TE2	0.00	0.00	8.72d ± 0.65	26.21c ± 1.26	30.65c ± 1.87	13.12E ± 3.25
TE3	0.00	0.00	8.63d ± 1.04	30.41b ± 1.15	34.61b ± 1.65	14.73D ± 4.04
TE4	0.00	0.00	7.16de ± 0.75	17.48d ± 1.02	30.12c ± 1.45	10.95F ± 3.67
S Control	0.00	0.00	19.40b ± 1.10	36.26a ± 1.76	45.61a ± 1.55	20.25B ± 3.85
T Control	0.00	0.00	13.76c ± 1.35	27.42c ± 1.20	44.28a ± 1.77	17.09C ± 4.52
Control	0.00	0.00	23.44a ± 1.00	37.49a ± 1.76	45.60a ± 1.19	21.31A ± 3.27
Mean (P) **	0.00	0.00	9.48C ± 1.61	21.49B ± 1.90	31.75A ± 2.58	
Significance	P0 = ND	P15 = ND	P30 = **	P45 = **	P60 = **	Treat. = **

Mean values (±SE) with different letters in each column are significant (at *p* ≤ 0.05); ND = not detected. ** = Significant at the 1% level of probability. Lowercase letters in the same column compare the average values of treatments (for the same storage period). Uppercase letters in a row Mean (P) represent a comparison of mean values between different periods (for all treatments). Uppercase letters in the last column represent a comparison of the mean values between different treatments (for all storage periods).

**Table 3 plants-12-03848-t003:** Effect of pre- and postharvest treatments on the fruit length (cm) of acid lime (*Citrus aurantifolia* swingle) fruits.

	Period (P; Days)					
Treatment	0	15	30	45	60	Mean (Treat.)
SE1	4.07a ± 0.25	3.90a ± 0.23	3.60a ± 0.26	3.73a ± 0.30	3.53a ± 0.29	3.77AB ± 0.24
SE2	4.10a ± 0.11	3.83a ± 0.12	4.10a ± 0.21	3.53a ± 0.22	3.47a ± 0.24	3.81A ± 0.36
SE3	4.03a ± 0.15	4.13a ± 015	3.87a ± 0.22	3.70a ± 0.22	3.73a ± 0.22	3.87A ± 0.38
SE4	3.83a–c ± 0.12	3.83a ± 0.12	3.70a ± 0.28	3.67a ± 0.32	3.23a ± 0.30	3.65BC ± 0.34
TE1	4.07a ± 0.15	3.80a ± 0.20	3.60a ± 0.26	3.47a ± 0.27	3.17a ± 0.31	3.62BC ± 0.29
TE2	3.83a–c ± 0.14	3.67a ± 0.15	3.63a ± 0.32	3.60a ± 0.22	3.58a ± 0.26	3.66BC ± 0.32
TE3	3.90ab ± 0.10	3.73a ± 0.15	3.60a ± 0.17	3.53a ± 0.18	3.20a ± 0.20	3.59CD ± 0.27
TE4	3.95ab ± 0.15	3.70a ± 0.10	3.67a ± 0.20	3.50a ± 0.24	3.47a ± 0.22	3.66BC ± 0.26
S Control	3.50c ± 0.24	3.47a ± 0.21	3.43a ± 0.15	3.50a ± 0.12	3.43a ± 0.23	3.47DE ± 0.37
T Control	3.63bc ± 0.21	3.57a ± 0.16	3.47a ± 0.15	3.43a ± 0.12	3.33a ± 0.21	3.49DE ± 0.35
Control	3.53c ± 0.12	3.47a ± 0.15	3.43a ± 0.21	3.40a ± 0.20	3.13a ± 0.25	3.39F ± 0.20
Mean (P) **	3.86A ± 0.24	3.74B ± 0.23	3.65B ± 0.25	3.54C ± 0.21	3.39D ± 0.28	
Significance	P0 = *	P15 = NS	P30 = NS	P45 = NS	P60 = NS	Treat. = **

Mean values (±SE) with different letters in each column are significant (at *p* ≤ 0.05); NS = the same letters are not significantly different (at *p* ≤ 0.05). * = Significant at the 0.05% level of probability. ** = Significant at the 1% level of probability. Lowercase letters in the same column compare the average values of treatments (for the same storage period). Uppercase letters in a row Mean (P) represent a comparison of the mean values between different periods (for all treatments). Uppercase letters in the last column represent a comparison of the mean values between different treatments (for all storage periods).

**Table 4 plants-12-03848-t004:** Effect of pre- and postharvest treatments on the fruit weight (g) of acid lime (*Citrus aurantifolia* swingle) fruits.

	Period (P; Days)					
Treatment	0	15	30	45	60	Mean (Treat.)
SE1	27.87b ± 0.42	25.94cd ± 0.70	25.62b ± 0.61	25.53b ± 0.71	24.22a ± 0.90	25.84C ± 1.55
SE2	28.72a ± 0.85	28.80a ± 0.60	27.77a ± 0.52	27.43a ± 0.48	22.23b ± 0.77	26.99A ± 3.20
SE3	27.71ab ± 0.50	27.09b ± 0.72	27.35a ± 0.59	25.34b ± 0.45	24.63a ± 1.00	26.41B ± 1.53
SE4	24.80fg ± 0.56	24.62f ± 0.61	24.42cd ± 0.70	22.99c ± 0.93	19.35c ± 1.10	23.23E ± 2.27
TE1	27.67cd ± 0.47	25.70c–e ± 0.70	23.83d ± 0.76	21.41d ± 1.02	17.47d ± 0.70	23.05E ± 3.67
TE2	27.00c ± 0.77	26.43bc ± 0.78	26.05b ± 0.76	25.50b ± 0.90	25.48a ± 0.46	26.09BC ± 2.53
TE3	25.93de ± 0.55	25.36d–f ± 0.70	25.07bc ± 0.73	23.17c ± 1.00	21.80b ± 0.70	24.27D ± 3.19
TE4	25.49ef ± 0.40	25.18ef ± 0.54	23.68d ± 0.56	21.22d ± 1.07	20.48c ± 0.75	23.19E ± 2.37
S Control	23.65g ± 0.60	23.27g ± 0.56	20.82e ± 0.87	20.66d ± 0.46	19.31c ± 0.95	21.53F ± 2.08
T Control	21.72h ± 0.45	21.40h ± 0.66	19.04f ± 0.71	18.91e ± 1.10	17.03d ± 0.94	19.62G ± 3.16
Control	18.42i ± 0.52	18.32i ± 0.40	17.20g ± 0.48	16.77f ± 0.72	13.37e ± 0.87	16.81H ± 3.54
Mean (P) **	25.26A ± 2.91	24.73B ± 2.19	23.70C ± 3.46	22.63D ± 3.11	20.49E ± 3.30	
Significance	P0 = **	P15 = **	P30 = **	P45 = **	P60 = **	Treat. = **

Mean values (±SE) with different letters in each column are significant (at *p* ≤ 0.05). ** = Significant at the 1% level of probability. Lowercase letters in the same column compare the average values of treatments (for the same storage period). Uppercase letters in a row Mean (P) a comparison of mean values between different periods (for all treatments). Uppercase letters in the last column compare the mean values between different treatments (for all storage periods).

**Table 5 plants-12-03848-t005:** Effect of pre- and postharvest treatments on the juice volume (cm^3^) of acid lime (*Citrus aurantifolia* S.) fruits.

	Period (P; Days)					
Treatments	0	15	30	45	60	Mean (Treat.)
SE1	12.17bc ± 0.40	13.00ab ± 0.43	11.00c ± 0.60	11.33de ± 0.27	10.33ab ± 0.52	11.57C ± 0.30
SE2	14.33a ± 0.46	13.67a ± 0.42	13.36a ± 0.34	13.00b ± 0.26	9.39bc ± 0.30	12.86A ± 0.43
SE3	13.93a ± 0.36	13.97a ± 0.26	13.37a ± 0.36	14.11a ± 0.18	9.33b–d ± 0.40	12.94A ± 0.50
SE4	11.00d ± 0.40	11.60c ± 0.33	11.69bc ± 0.36	12.00cd ± 0.27	7.67ef ± 0.52	10.79D ± 0.45
TE1	12.67b ± 0.40	13.00ab ± 0.42	12.33ab ± 0.34	11.00ef ± 0.30	8.53c–e ± 0.73	11.52C ± 0.48
TE2	11.67cd ± 0.46	8.78d ± 0.44	8.67d ± 0.60	10.31f ± 0.16	8.31de ± 0.50	9.554E ± 0.38
TE3	13.03b ± 0.46	11.33c ± 0.46	12.33a–c ± 0.33	12.67bc ± 0.27	11.67a ± 0.25	12.21B ± 0.32
TE4	11.70cd ± 0.40	12.00bc ± 0.48	11.33bc ± 0.56	10.69ef ± 0.34	9.31b–d ± 0.50	11.01D ± 0.30
S Control	11.11d ± 0.30	11.11c ± 0.37	9.67d ± 0.69	8.98g ± 0.30	6.53f ± 0.35	9.48E ± 0.47
T Control	11.00d ± 0.33	11.17c ± 0.28	9.00d ± 0.40	7.80h ± 0.32	7.33ef ± 0.29	9.26E ± 0.43
Control	9.67e ± 0.41	7.67d ± 0.24	6.00e ± 0.60	5.67i ± 0.27	4.98g ± 0.50	6.80F ± 0.48
Mean (P) **	12.03A ± 0.65	11.57B ± 0.44	10.79C ± 0.70	10.69C ± 0.43	8.54D ± 0.63	
Significance	P0 = **	P15 = **	P30 = **	P45 = **	P60 = **	Treat. = **

Mean values (±SE) with different letters in each column are significant (at *p* ≤ 0.05). ** = Significant at the 1% level of probability. Lowercase letters in the same column compare the average values of treatments (for the same storage period). Uppercase letters in a row Mean (P) a comparison of mean values between different periods (for all treatments). Uppercase letters in the last column compare the mean values between different treatments (for all storage periods).

**Table 6 plants-12-03848-t006:** Effect of pre- and postharvest treatments on the TSS of acid lime (*Citrus aurantifolia* swingle) fruits.

	Period (P; Days)					
Treatment	0	15	30	45	60	Mean (Treat.)
SE1	7.87a ± 0.23	7.63a ± 0.33	8.17a ± 0.33	7.67ab ± 0.18	8.27a ± 0.42	7.92AD ± 0.39
SE2	7.70a ± 0.40	7.83a ± 0.38	7.80bc ± 0.34	7.77ab ± 0.41	7.87a ± 0.42	7.79BC ± 0.34
SE3	7.77a ± 0.37	7.80a ± 0.40	7.63c ± 0.38	7.23b ± 0.28	8.00a ± 0.38	7.69DE ± 0.34
SE4	7.67a ± 0.33	8.17a ± 0.31	7.57c ± 0.43	7.57ab ± 0.32	7.80a ± 0.45	7.75CE ± 0.35
TE1	7.53a ± 0.32	7.97a ± 0.33	7.60c ± 0.22	8.20ab ± 0.48	8.00a ± 0.50	7.86BE ± 0.38
TE2	7.93a ± 0.35	8.00a ± 0.43	8.07a ± 0.35	8.20ab ± 0.38	8.07a ± 0.46	7.99AB ± 0.38
TE3	7.73a ± 0.38	7.50a ± 0.32	7.57c ± 0.33	8.50a ± 0.26	8.63a ± 0.43	7.99AB ± 0.28
TE4	7.97a ± 0.22	8.20a ± 0.22	7.93bc ± 0.28	8.43a ± 0.32	8.33a ± 0.40	8.17A ± 0.32
S Control	8.07a ± 0.32	8.07a ± 0.54	7.77bc ± 0.12	7.50ab ± 0.29	7.30a ± 0.40	7.74CE ± 0.38
T Control	7.73a ± 0.41	7.80a ± 0.21	7.20d ± 0.18	7.53ab ± 0.17	7.73a ± 0.47	7.60E ± 0.31
Control	7.10a ± 0.35	6.87b ± 0.33	6.67e ± 0.18	7.23b ± 0.28	6.83a ± 0.38	6.94F ± 0.33
Mean (P) *	7.63B ± 0.46	7.80AB ± 0.30	7.63B ± 0.42	7.80AB ± 0.35	7.89A ± 0.39	
Significance	P0 = NS	P15 = *	P30 = **	P45 = *	P60 = NS	Treat. = **

Mean values (±SE) with different letters in each column are significant (at *p* ≤ 0.05); NS = the same letters are not significantly different (at *p* ≤ 0.05). * = Significant at the 0.05% level of probability. ** = Significant at the 1% level of probability. Lowercase letters in the same column compare the average values of treatments (for the same storage period). Uppercase letters in a row Mean (P) a comparison of mean values between different periods (for all treatments). Uppercase letters in the last column compare the mean values between different treatments (for all storage periods).

**Table 7 plants-12-03848-t007:** Effect of pre- and postharvest treatments on the titratable Acidity (TA) of acid lime (*Citrus aurantifolia* S.) fruits.

	Period (P; Days)					
Treatment	0	15	30	45	60	Mean (Treat.)
SE1	5.97c ± 0.11	6.93a–d ± 0.18	6.93a ± 0.15	5.55cd ± 0.22	5.61a ± 0.20	6.23B ± 0.29
SE2	6.83a–c ± 0.15	7.89a–c ± 0.21	7.10a ± 0.30	6.19a–c ± 0.15	5.50a ± 0.25	6.70B ± 0.22
SE3	6.53bc ± 0.18	6.93a–d ± 0.26	6.93a ± 0.23	5.97bc ± 0.26	5.50a ± 0.21	6.37B ± 0.27
SE4	6.74a–c ± 0.22	6.53cd ± 0.21	7.68a ± 0.26	5.12cd ± 0.29	6.02a ± 0.32	6.42B ± 0.24
TE1	7.23a–c ± 0.52	6.63b–d ± 0.26	7.17a ± 0.30	4.80d ± 0.21	5.44a ± 0.29	6.25B ± 0.29
TE2	6.98a–c ± 0.15	7.17a–d ± 0.29	7.68a ± 0.26	5.18cd ± 0.29	4.48a ± 0.27	6.30B ± 0.38
TE3	6.72a–c ± 0.32	5.87d ± 0.26	7.04a ± 0.30	6.10bc ± 0.29	6.40a ± 0.21	6.43B ± 0.29
TE4	7.04a–c ± 0.27	7.62a–c ± 0.24	7.17a ± 0.21	5.89b–d ± 0.30	5.44a ± 0.29	6.63B ± 0.20
S Control	7.96ab ± 0.22	8.30a ± 0.34	7.19a ± 0.27	7.25a ± 0.23	6.44a ± 0.26	7.43A ± 0.31
T Control	7.89ab ± 0.21	8.17ab ± 0.23	7.06a ± 0.20	7.02ab ± 0.26	6.85a ± 0.25	7.40A ± 0.28
Control	8.04a ± 0.24	8.23ab ± 0.27	7.13a ± 0.20	6.89ab ± 0.23	6.72a ± 0.25	7.40A ± 0.20
Mean (P) **	7.08A ± 0.23	7.30A ± 0.28	7.19A ± 0.20	6.00B ± 0.26	5.86B ± 0.25	
Significance	P0 = *	P15 = *	P30 = NS	P45 = *	P60 = NS	Treat. = **

Mean values (±SE) with different letters in each column are significant (at *p* ≤ 0.05); NS = the same letters are not significantly different (at *p* ≤ 0.05). * = Significant at the 0.05% level of probability. ** = Significant at the 1% level of probability. Lowercase letters in the same column compare the average values of treatments (for the same storage period). Uppercase letters in a row Mean (P) a comparison of mean values between different periods (for all treatments). Uppercase letters in the last column compare the mean values between different treatments (for all storage periods).

**Table 8 plants-12-03848-t008:** Effect of pre- and postharvest treatments on the vitamin C content of acid lime (*Citrus aurantifolia* S.) fruits.

	Period (P; Days)					
Treatment	0	15	30	45	60	Mean (Treat.)
SE1	56.53ab ± 2.82	54.19c ± 1.61	65.33a ± 2.10	47.73cd ± 1.86	39.47b ± 1.93	52.65A ± 2.70
SE2	62.93a ± 2.87	59.41bc ± 1.90	57.81bc ± 2.40	44.80de ± 1.85	49.60a ± 2.26	54.91A ± 2.39
SE3	55.73ab ± 2.94	54.51c ± 1.95	53.33c ± 2.60	52.80ab ± 2.57	38.40b ± 2.26	50.95A ± 2.54
SE4	56.85ab ± 2.76	64.00ab ± 2.40	46.93d ± 1.60	50.13bc ± 2.87	37.33b ± 2.10	51.05A ± 3.19
TE1	49.65b ± 2.87	46.13d ± 1.91	46.24d ± 1.80	41.07e ± 1.46	38.93b ± 1.70	44.41B ± 2.28
TE2	55.25ab ± 2.00	56.85c ± 1.90	57.33bc ± 2.03	44.80de ± 1.86	37.60b ± 1.41	50.37A ± 2.74
TE3	62.67a ± 1.11	66.29a ± 1.64	59.15b ± 1.85	54.93ab ± 2.41	35.20bc ± 1.40	55.65A ± 2.99
TE4	62.19a ± 1.91	58.13c ± 1.84	62.08ab ± 1.20	55.47a ± 2.10	35.20bc ± 2.95	54.61A ± 3.13
S Control	50.93b ± 2.90	42.13d ± 1.30	36.80f ± 1.40	29.33g ± 2.77	27.73d ± 2.61	37.39C ± 2.94
T Control	33.65c ± 2.10	27.73f ± 1.95	37.33f ± 1.45	31.20g ± 2.20	29.33cd ± 2.62	31.85D ± 1.83
Control	34.88c ± 2.80	34.13e ± 1.12	41.07ef ± 2.66	36.27f ± 1.78	26.13d ± 2.70	34.50CD ± 2.05
Mean (P) **	52.84A ± 2.09	51.23A ± 2.41	51.22A ± 1.98	44.41B ± 1.93	35.90C ± 1.41	
Significance	P0 = **	P15 = **	P30 = **	P45 = **	P60 = **	Treat. = **

Mean values (±SE) with different letters in each column are significant (at *p* ≤ 0.05). ** = Significant at the 1% level of probability. Lowercase letters in the same column compare the average values of treatments (for the same storage period). Uppercase letters in a row Mean (P) a comparison of the means values between different periods (for all treatments). Uppercase letters in the last column compare the mean values between different treatments (for all storage periods).

## Data Availability

The datasets generated during and/or analyzed during the current study are available from the corresponding author upon reasonable request.

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
