# Peer review of "Potassium Spraying Preharvest and Nanocoating Postharvest Improve the Quality and Extend the Storage Period for Acid Lime (Citrus aurantifolia Swingle) Fruits"

_plants, 2023, doi:10.3390/plants12223848_

Round 1
Reviewer 1 Report
Comments and Suggestions for Authors
[Plants] Manuscript ID: plants-2630370 - Review Report
Dear Editor,
the topic of the manuscript entitled “Potassium spraying at pre-harvest and nanocoating at post-harvest improve the quality and extend storage period for acid lime (Citrus aurantifolia) fruits” is very interesting. It aimed to demonstrate the enhancing effect of pre-harvest spraying with potassium both in mineral and organic form, in addition to post-harvest dipping of fruits in some edible coatings on the quality and storability of acid lime fruits.
In this work, a green ecofriendly method was used for the biosynthesis of Zinc oxide nanoparticles (ZnONPs) using acid lime peel waste extract, incorporated into Carboxymethyl cellulose (CMC) matrix to formulate an edible nanocoating for acid lime fruits.
In general, the quality of the study is good. The abstract is well articulated and summarizes the content of the article well. In my opinion this manuscript should be subjected to a minor revision before being considered for publication in the Plants journal. Some points should be clarified or discussed. I suggest the Authors consider the following points:
Statistical analysis and discussion of the results referring individually to each storage period would be desirable (One-way ANOVA).
It would be preferable to assign a bulleted list to the titles of the sub-paragraphs, otherwise the reading is confusing. A bulleted list relating to the paragraph number is required.
The correspondence and appropriateness of the references throughout the article must be verified. Some references seem off-center, others are missing.
Please consider the following specific suggestions:
Line 23: For further precision and completeness, it should be mentioned how the control was treated in the pre-harvest phase.
Line 11: Add Swingle or S. to the Citrus aurantifolia nomenclature.
Line 91-92: Insert a row between 91 and 92 lines.
Lines 146-148: Standardize the font.
Lines 156-202: Sub-paragraph titles must be standardized and made less confusing. A bulleted list relative to the paragraph number would probably make a difference.
Line160: Please correct the title of the sub-paragraph.
Lines 198 and 200. The AOAC (2012) standard methods are not mentioned in the References.
Line 206: Please write “p ≤ 0.05” instead of “p = 0.5”.
Lines 227-232. Figure 4 is not cited in the text.
Lines 254 and following: The numbering of the subparagraphs is not very clear. A bulleted list relating to the paragraph number is required.
Lines 323-327: Table 4 is not cited in the text.
Lines 471 and 475: Sajid et al., 2022 is not mentioned in the References.
Line 484: Alva et al., 2006 is not mentioned in the References.
Line 486: There is a typo (26 ACS).
Line 484: Prasad et al., 2015 is not mentioned in the References.
Line 504: The reference number for Alva et al., 2006 is missing.
Tables and Figures:
- Please cite the Duncan post hoc test at p ≤ 0.05 in the caption of tables and figures.
- Table 1 is not very clear. Please reconfigure.
- Table 4 is not cited in the text.
- In tables 2 to 8, the statistical analysis should also be carried out for each storage period (one way ANOVA). Each column should be statistically analyzed individually, and the discussion should be also related to these results.
- Figure 4 is not cited in the text.
- As with tables 2 to 8, in figures 7 to 11 the statistical analysis should be carried out for each storage period (one way ANOVA).
References:
- The AOAC (2012) standard methods are not mentioned in the References.
- Sajid et al., 2022; Alva et al., 2006; and Prasad et al., 2015, are not mentioned properly.

Author Response
Reviewer #1
Dear Editor,
the topic of the manuscript entitled “Potassium spraying at pre-harvest and nanocoating at post-harvest improve the quality and extend storage period for acid lime (Citrus aurantifolia) fruits” is very interesting. It aimed to demonstrate the enhancing effect of pre-harvest spraying with potassium both in mineral and organic form, in addition to post-harvest dipping of fruits in some edible coatings on the quality and storability of acid lime fruits.
In this work, a green ecofriendly method was used for the biosynthesis of Zinc oxide nanoparticles (ZnONPs) using acid lime peel waste extract, incorporated into Carboxymethyl cellulose (CMC) matrix to formulate an edible nanocoating for acid lime fruits.
In general, the quality of the study is good. The abstract is well articulated and summarizes the content of the article well. In my opinion this manuscript should be subjected to a minor revision before being considered for publication in the Plants journal.
Response: The authors deeply thank the reviewer for his/her revision and suggested comments that aimed to enhance and improve our work.
Some points should be clarified or discussed. I suggest the Authors consider the following points:
Statistical analysis and discussion of the results referring individually to each storage period would be desirable (One-way ANOVA).
Response: The authors completely agree with the reviewer's notice and according to this comment the statistical analysis was performed for each storage period.
It would be preferable to assign a bulleted list to the titles of the sub-paragraphs, otherwise, the reading is confusing. A bulleted list relating to the paragraph number is required.
Response: The bulleted numbering was used according to the reviewer's advice.
The correspondence and appropriateness of the references throughout the article must be verified. Some references seem off-center, others are missing.
Response: References were revised carefully and now they are well-cited.
Please consider the following specific suggestions:
Line 23: For further precision and completeness, it should be mentioned how the control was treated in the pre-harvest phase.
Response: Control trees were sprayed with distilled water.
Line 11: Add Swingle or S. to the Citrus aurantifolia nomenclature.
Response: Done, written in red color.
Line 91-92: Insert a row between 91 and 92 lines.
Response: done.
Lines 146-148: Standardize the font.
Response: done, in red color.
Lines 156-202: Sub-paragraph titles must be standardized and made less confusing. A bulleted list relative to the paragraph number would probably make a difference.
Response: Done.
Line160: Please correct the title of the sub-paragraph.
Response: Corrected.
Lines 198 and 200. The AOAC (2012) standard methods are not mentioned in the References.
Response: Done and added to references.
Line 206: Please write “p ≤ 0.05” instead of “p = 0.5”.
Response: done.
Lines 227-232. Figure 4 is not cited in the text.
Response: Now, Figure 4 is cited in the text and explained.
Lines 254 and following: The numbering of the subparagraphs is not very clear. A bulleted list relating to the paragraph number is required.
Response: Corrected.
Lines 323-327: Table 4 is not cited in the text.
Response: done and Table 4 citation is inserted in the text.
Lines 471 and 475: Sajid et al., 2022 is not mentioned in the References.
Response: done and added to references.
Line 484: Alva et al., 2006 is not mentioned in the References.
Response: done and added to references
Line 486: There is a typo (26 ACS).
Response: References are corrected in the manuscript.
Line 484: Prasad et al., 2015 is not mentioned in the References.
Response: Done and added to references
Line 504: The reference number for Alva et al., 2006 is missing.
Response: References are corrected in the manuscript.
Tables and Figures:
- Please cite the Duncan post hoc test at p ≤ 0.05 in the caption of tables and figures.
Response: Done.
- Table 1 is not very clear. Please reconfigure.
Response: Modified.
- Table 4 is not cited in the text.
Response: Done and added in the text.
- In tables 2 to 8, the statistical analysis should also be carried out for each storage period (one way ANOVA). Each column should be statistically analyzed individually, and the discussion should be also related to these results.
Response: Done and statistical analyses were performed for each storage period.
- Figure 4 is not cited in the text.
Response: Now, it is cited and explained in the main text.
- As with tables 2 to 8, in figures 7 to 11 the statistical analysis should be carried out for each storage period (one way ANOVA).
Response: Done and statistical analyses were performed for each storage period.
References:
- The AOAC (2012) standard methods are not mentioned in the References.
Response: Done and added to references
- Cunniff, P.; Washington, D. Official methods of analysis of AOAC international. J. AOAC Int 1997, 80, 127A.
- Sajid et al., 2022; Alva et al., 2006; and Prasad et al., 2015, are not mentioned properly.
Response: Done and added to references.
Reviewer 2 Report
Comments and Suggestions for Authors
Comments on paper title “Potassium spraying at pre-harvest and nanocoating at post-harvest improve the quality and extend storage period for acid lime (Citrus aurantifolia) fruits”.
The article has a lot of data but the type of analyses is very simple, which doesn't allow us to draw more in-depth conclusions.
The design is not well explained and some codes are not identified, which makes it impossible to understand.
Some techniques are not well explained and discussed.
in general, the article is poor and has many formal gaps.
The subchapters and other things are not formatted according to the journal's guidelines.
Only some examples:
The scientific names must be in italic revise in all document
Line 61: “…linkages. [11,12].” Mut be “…linkages [11,12].”
Is not clear the Preharvest treatment, could the authors explain better or made a scheme
Figure 1 The unit of temperature is not correct 2 decimal numbers for the unit are not necessary
The unit ºC in incorrect. A space between number and unit is necessary. Revise the units in all document there are a lot of mistakes.
Line 120: preprepared ??
”... doped with ZnONPs... “ revise sentence
146 - 148 different formats
Line 200 miss reference
In material and methods, the FTIR analysis is not explained and in not discussed in an appropriate way. Line 218 a 22 is incorrect
The legend of the figures is not complete
Appears some codes notes not explain which makes difficult to understand the discussion
Etc.
Author Response
Reviewer #2
Comments on paper title “Potassium spraying at pre-harvest and nanocoating at post-harvest improve the quality and extend storage period for acid lime (Citrus aurantifolia) fruits”.
The article has a lot of data but the type of analyses is very simple, which doesn't allow us to draw more in-depth conclusions.
Response: The authors deeply agree with the reviewer. Additionally, the authors would like to thank the reviewer for his/her careful revision of our article to improve and enhance the article's performance as much possible as. According to this comment, the whole article was revised carefully to avoid something like this and reformulated to make it easy to follow the story.
The design is not well explained and some codes are not identified, which makes it impossible to understand.
Response: The explanation of all parts was revised and reformulated as well as the codes were revised and corrected according to your notice.
Some techniques are not well explained and discussed.
Response: All techniques were revised well and represented to be clear.
in general, the article is poor and has many formal gaps.
Response: The whole article was revised carefully to avoid any gaps.
The subchapters and other things are not formatted according to the journal's guidelines.
Response: corrected.
Only some examples:
The scientific names must be in italic revise in all document
Response: Done, corrected in red color.
Line 61: “…linkages. [11,12].” Mut be “…linkages [11,12].”
Response: Corrected.
Is not clear the Preharvest treatment, could the authors explain better or made a scheme
Response: done (in materials and methods section; lines 94 -103) and (in discussion section; lines 480 482).
Figure 1 The unit of temperature is not correct 2 decimal numbers for the unit are not necessary
Response: Corrected.
The unit ºC in incorrect. A space between number and unit is necessary. Revise the units in all document there are a lot of mistakes.
Response: Corrected.
Line 120: preprepared ??
Response: corrected.
”... doped with ZnONPs... “ revise sentence
Response: Rephrased.
146 - 148 different formats
Response: Done, in red color.
Line 200 miss reference
Response: Done and added to references
In material and methods, the FTIR analysis is not explained and in not discussed in an appropriate way. Line 218 a 22 is incorrect
Response: The FTIR part in methodology, results and discussion were reformulated to be clear.
The legend of the figures is not complete
Response: All figures ligands were revised carefully and modified.
Appears some codes notes not explain which makes difficult to understand the discussion
Etc.
Response: Corrected
Reviewer 3 Report
Comments and Suggestions for Authors
In article "Potassium spraying at pre-harvest and nanocoating at post-harvest improve the quality and extend storage period for acid lime (Citrus aurantifolia) fruits" must be 1.clearly present the semification of abreviation ex in abstract L19 ZnONPs, E1,E2...
2,In table all table you calculate mean of value at 0,15, 30, 45 and 60 days, for what? Also you have some letter after the value -semnification
Author Response
Reviewer #3
Comments and Suggestions for Authors
In article "Potassium spraying at pre-harvest and nanocoating at post-harvest improve the quality and extend storage period for acid lime (Citrus aurantifolia) fruits" must be 1.clearly present the semification of abreviation ex in abstract L19 ZnONPs, E1,E2...
Response: The authors would like to acknowledge the reviewer's comments and effort. According to this comment, the whole article was deeply and carefully revised to avoid the weakness points that the reviewer mentioned and the article is now presented in a carefully revised form.
2,In table all table you calculate mean of value at 0,15, 30, 45 and 60 days, for what? Also you have some letter after the value -semnification
Response: We wanted to see the general trend of cold storage periods, and a separate statistical analysis was performed to compare the periods.
Round 2
Reviewer 2 Report
Comments and Suggestions for Authors
The authors made significant modification in the paper that made him better and more understanding, So in my opinion the paper could be accepted for publication in the Plants Journal
Only some minor improvements:
in cm−1 the -1 must be superscript
line 364: kg.cm−2 must be kg cm−2 or in othe whay, but all with the same format, appear cm3/L.
Please carefully revise all unite write because there are many different styles. Please Uniformize
Author Response
The whole manuscript was revised.
The requested corrections were addressed carefully.